# A Ru-Doped VTi$_{2.6}$O$_{7.2}$ Anode with High Conductivity for Enhanced Sodium Storage

Guangwan Zhang [1,2], Chunhua Han [1,2,*], Kang Han [1], Jinshuai Liu [1,2], Jinghui Chen [1,2], Haokai Wang [1,2], Lei Zhang [2,3,*] and Xuanpeng Wang [2,3,*]

1    School of Materials Science and Engineering, Hainan Institute, Wuhan University of Technology, Wuhan 430070, China
2    Hainan Institute, Wuhan University of Technology, Sanya 572000, China
3    Department of Physical Science & Technology, School of Science, Wuhan University of Technology, Wuhan 430070, China
*    Correspondence: hch5927@whut.edu.cn (C.H.); zhanglei1990@whut.edu.cn (L.Z.); wxp122525691@whut.edu.cn (X.W.)

**Abstract:** Sodium-ion batteries (SIBs) are considered a potential replacement for lithium-ion batteries in the area of low-cost large-scale energy storage. Due to its low operating voltage, high capacity, non-toxicity and low production cost, titanium dioxide is now among the anode materials under investigation and shows the most promise. However, its poor electrical conductivity is one of the main reasons limiting its large-scale application. Herein, we designed a ruthenium-doped anatase-type VTi$_{2.6}$O$_{7.2}$ ultrafine nanocrystal (Ru-VTO). As the anode of SIBs, Ru-VTO delivers a high specific capacity of 297 mAh g$^{-1}$ at 50 mA g$^{-1}$, a long cycle life of 2000 cycles and a high rate capability (104 mAh g$^{-1}$ at 1000 mA g$^{-1}$). The excellent performance may be related to the solid-solution interatomic interactions and the enhanced conductivity after ruthenium doping. These studies demonstrate the potential of Ru-VTO as an anode material for advanced SIBs.

**Keywords:** titanium-based material; ru doping; high conductivity; anode; sodium-ion battery





## 1. Introduction

With severe global energy and environmental problems, it is vitally necessary to build ecologically friendly and effective energy storage devices [1,2]. Due to their plentiful reserves and low prices, sodium-ion batteries (SIBs) have received a lot of attention as a substitute for lithium-ion batteries in recent years [3–8]. Although the electrochemical reaction mechanism of the two battery electrode materials is similar, applying already-developed LIB electrode materials to SIBs is not always effective due to the large radius of Na$^+$ ions (0.102 nm for Na$^+$ and 0.076 nm for Li$^+$) [9–11]. Therefore, the development of high-performance anode materials for SIBs is an important task and the key to large-scale applications. Among anode materials for SIBs, embedded/detached materials, such as hard carbon [12], molybdenum-based materials [13] and titanium-based materials [14], are considered strong contenders for SIB anode materials due to their stable structural properties, excellent kinetic characteristics and a suitable sodium ion insertion potential [15,16].

TiO$_2$ has attracted significant scientific interest as an enhanced salt storage system because of its high theoretical capacity (335 mAh g$^{-1}$), acceptable insertion potential (0.7 V), negligible volume expansion and low cost [17–22]. However, the low intrinsic electron conductivity and application of SIBs are still constrained by ionic conductivity [23–25]. Currently, several solutions have been presented to improve the electrical conductivity of TiO$_2$ anodes in conjunction with rigorous structural designs, so as to attain excellent sodium storage performance.

The incorporation of the guest element into the matrix to form a solid solution can effectively lower the energy potential barrier for sodium ion diffusion, increasing conductivity. In addition, the composition of solid solutions can be more easily adjusted, and their

easy adjustment properties further lead to a diversity of active materials. Previous research has shown that, with the development of substituted solid solutions, electrochemical performance can be effectively enhanced by altering interatomic interactions [26,27]. Sheng et al. introduced vanadium with a high electrochemical activity into the $TiO_2$ lattice to form substituted solid solutions with titanium, and when applied to a mixed magnesium–lithium battery a capacity of 265.2 mAh $g^{-1}$ was provided at the current density of 50 mA $g^{-1}$ with a good performance [28]. However, solid solutions still suffer from low conductivity when used as anode materials for SIBs. Numerous investigations have demonstrated that heteroatom doping has significant potential benefits as a very effective means to improve the electrochemical properties of metal oxides. The electrical conductivity of $TiO_2$ is increased through doping by redistributing charge around the doping sites [29–32]. For example, M. M. S. Sanad reported on mesoporous $TiO_2$ introduced by noble metal nanoparticles with enhanced electrical conductivity [33]. Guo et al. found that, through Ru doping, the resistance can be effectively reduced while increasing the electron transfer rate, and meanwhile the material structural stability is improved [34]. Thus, enhancing the electronic conductivity of solid solutions through Ru doping can be attempted.

Herein, a new-type ruthenium-doped anatase type $VTi_{2.6}O_{7.2}$ ultrafine nanocrystal (Ru-VTO) anode material was designed and prepared by a straightforward hydrothermal synthesis procedure, followed by annealing, for the first time. By incorporating vanadium into the $TiO_2$ lattice to form a replacement solid solution and simultaneously doping the above solution with ruthenium, the high electrochemical activity of the solid solution is ensured while maintaining its structural stability. The well-designed Ru-VTO exhibits an excellent capacity of 297 mAh $g^{-1}$, as well as a high rate capability and cycling stability (104 mAh $g^{-1}$ at 1000 mA $g^{-1}$ after 2000 cycles) when used as SIB anode material.

## 2. Materials and Methods

### 2.1. Sample Preparation

There was no additional purification required because all chemicals and reagents were analytical grade. V(IV)O(acac)$_2$ (99%), $K_2TiO(C_2O_4)_2$ (99%), ruthenium chloride (RuCl$_3$, 99%) and hydrogen peroxide solution (30%) were purchased from Aladdin (Shanghai, China) and Macklin (Shanghai, China).

Ru-VTO was synthesized as follows: in 70 mL of deionized water, 2 mmol of V(IV)O(acac)$_2$ blue-green crystal and 2 mmol of $K_2TiO(C_2O_4)_2$ white crystalline powder were dissolved by vigorously swirling for about 30 min. Then, to the aforementioned solution, ruthenium chloride solution (0.6 mL, 0.1 mol $L^{-1}$) and hydrogen peroxide solution (5 mL, 30%) were added. The resultant solution was then put into a stainless steel autoclave lined with Teflon, which was heated to 200 °C for 10 h. The products were regularly rinsed in ethanol and deionized water, and then dried for 12 h at 70 °C in a vacuum. The as-prepared samples were annealed at 450 °C for 3 h in an Ar atmosphere.

The samples prepared through the hydrothermal method followed by annealing with RuCl$_3$ are identified as Ru-VTO, and those without RuCl$_3$ after annealing are identified as VTO.

### 2.2. Electrochemical Measurements

The half 2016 coin-type cells with sodium metal as the counter electrode were used to investigate the electrochemical performance. Working electrodes were created by combining active components (VTO and Ru-VTO), acetylene black, polyvinylidene fluoride and solvent N-methyl-2-pyrrolidinone at a weight ratio of 7:2:1. The active mass loading range of electrodes is from 1.4 to 1.6 mg $cm^2$. A Whatman glass fiber (GF/A) was employed as the separator, and meanwhile propylene carbonate, fluoroethylene carbonate (95:5) and propylene carbonate were all mixed together to form a solution containing 1M NaClO$_4$. In a LAND CT2001A (Wuhan, China) testing system, galvanostatic charge–discharge experiments were carried out within 0.005–3 V (against Na/Na$^+$). An electrochemical workstation Autolab PGSTAT 302 (Herisau, Switzerland) was used to acquire cyclic voltammetry (CV)

curves and electrochemical impedance spectra (EIS). A GITT test was conducted using 20 mA g$^{-1}$ current pulses, lasting for a 10 min and a 30 min rest period.

### 2.3. Characterizations

A Bruker D8 Discover (Billerica, MA, USA) X-ray diffractometer with a planar detector was used in still mode to capture the XRD signals every 120 s. A JEOL-7100F (Tokyo, Japan) microscope was used to gather SEM pictures and energy dispersive spectrometer (EDS) data at an acceleration voltage of 15 kV. An ESCALAB 250Xi (Waltham, MA, USA) apparatus was used to conduct X-ray photoelectron spectroscopy (XPS).

## 3. Results

The overall fabrication process of Ru-VTO is briefly illustrated in Figure 1a. First of all, based on the previous work [28], ruthenium-doped VTi$_{2.6}$O$_{7.2}$ precursors were obtained by adding ruthenium chloride (RuCl$_3$) followed by a hydrothermal reaction. Indirect evidence of the effective introduction of ruthenium was obtained by observing that the color of the precursor powder changed from dark green to black-brown following the ruthenium doping. Figure 1b shows the XRD profiles of VTi$_{2.6}$O$_{7.2}$ and Ru-VTO. The main diffraction peaks of VTO matched well with the standard anatase TiO$_2$ (JCPDS No. 01-084-1285), which is consistent with the previous report [28], suggesting the pure phase of VTi$_{2.6}$O$_{7.2}$ and that the crystal structure is an anatase structure. The prepared Ru-VTO crystallized well, exhibiting the standard diffraction peaks of standard anatase TiO$_2$ with a slightly lower crystallinity compared to that of VTi$_{2.6}$O$_{7.2}$, but the diffraction peaks remained consistent. Additionally, the XRD profile (Figure S2) of the Ru-VTO precursor was evaluated, and its very poor crystallinity was shown by its very faint peak intensity. We calculated the crystallite size of Ru-VTO to be 10.66 nm by applying the Scherrer equation (Equation (S1), see Supplementary Materials) to the XRD profile [35]. The morphology and microstructure of Ru-VTO were characterized through SEM, and as is seen in Figures 1c and S1a the sample consists of many micrometer blocks with rough surfaces and a large number of nanoparticles wrapped around them. The SEM of VTO is seen in Figure S1b. The elemental distribution of Ru-VTO was further observed through EDS-mapping, and the results revealed that V, Ti, O and Ru were consistently distributed in the sample (Figures 1d, S12 and S13). According to EDS data, Ru has been effectively doped into the VTO sample.

The redox state of V, Ti and Ru in the Ru-VTO sample was investigated by measuring XPS. In Figure 2a, the signal of Ru was observed in the Ru-VTO sample and the successful incorporation of Ru into VTO was further demonstrated. The peaks at 517.54 eV and 524.74 eV in the V 2p nuclear level spectrum (Figure 2b) were connected to V 2p$_{3/2}$ and V 2p$_{1/2}$ of V$^{4+}$, respectively [36,37]. The peaks at 458.92 eV and 467.72 eV in the Ti 2p nuclear level spectrum (Figure 2c) were Ti 2p$_{3/2}$ and Ti 2p$_{1/2}$ of Ti$^{4+}$, respectively [38,39]. The existence of Ru$^{4+}$ was attributed to the presence of the Ru 3d$_{5/2}$ band at 281.0 eV (Figure 2d) [40]. The XPS spectra of O 1s for TiO$_2$, VTO and Ru-VTO are shown in Figure S11. Based on an elemental content analysis of XPS (Table S1), it could be concluded that the atomic content of Ru was 1.54%.

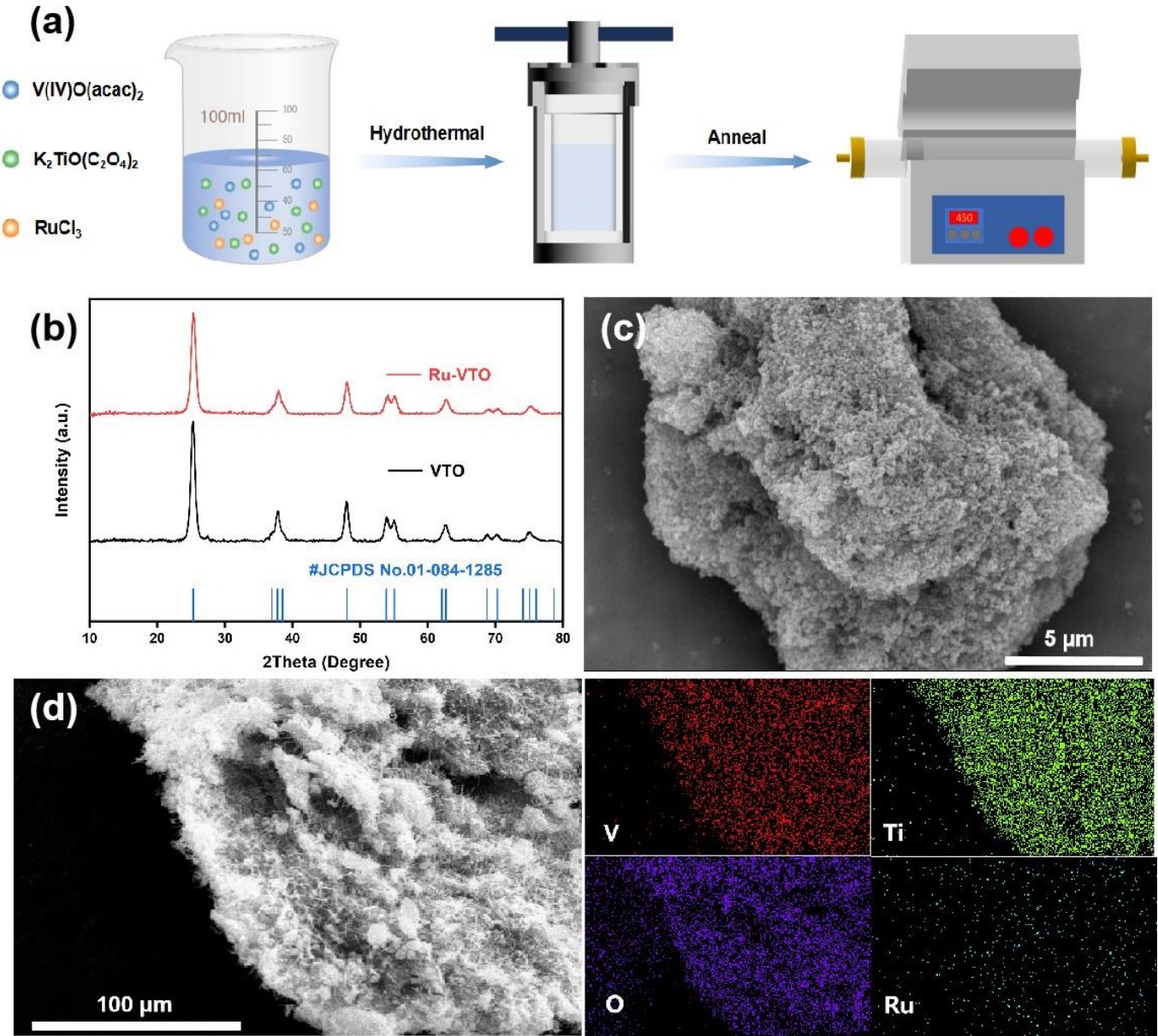

**Figure 1.** (**a**) Schematic illustration of the typical synthetic process of Ru-VTO. (**b**) XRD comparison of Ru-VTO and VTO. (**c**) Scanning electron microscopy (SEM) images of Ru-VTO. (**d**) Large-scale area EDS-mapping of Ru-VTO.

Cyclic voltammetry was used to examine the sodium storage behavior of Ru-VTO (CV). The strong cathodic peaks at about 1.1 V and 1.8 V disappeared after the initial scan, as is shown in Figure 3a, which was ascribed to solid electrolyte interphase (SEI) layer development and irreversible sodium intercalation [41,42]. A couple of reduction/oxidation peaks (0.913/0.690 V) with a peak deviation (ΔE) of 0.223 V were visible following the preliminary scan of the almost-overlapping CV curves of Ru-VTO, which contrasted well with that of VTO (ΔE: 0.424 V) (Figure S6). The enhanced electronic conductivity and diffusion kinetics brought about through Ru-doping were likely responsible for the decreased overpotential of Ru-VTO. Figure 3b shows the first charge/discharge curves of Ru-VTO and VTO at 50 mA g$^{-1}$. Galvanostatic charge and discharge voltage profiles of Ru-VTO and VTO under different current densities are seen in Figure S5a,b. The SEI layers and certain secondary reactions (irreversible reactions) between electrolyte and electrodes were observed to coincide with a bright outflow plateau with a slope extending from 1.31 to

0.9 V [43–46]. The original discharge/charge ratio capacity of Ru-VTO and VTO anodes was 602/303 and 600/244 mAh g$^{-1}$, corresponding to the initial Coulomb efficiency of 50.3% and 40.6%. The irreversible breakdown of the electrolyte, the restricted reversibility of the sodalization/desodalization process and other secondary effects all contributed to the initial capacity reduction. Apparently, Ru-VTO delivered a higher reversible capacity and initial Coulomb efficiency than the VTO sample, which was mostly ascribed to the enhanced electronic conductivity and diffusion kinetics brought forth via ruthenium doping. According to the cycling performance of Ru-VTO, VTO and pure TiO$_2$ at 50 mA g$^{-1}$ (Figures 3c and S8), Ru-VTO had a higher discharge capacity and more excellent cycling stability, while VTO and pure TiO$_2$ exhibited poor cycling performance. It can be found that the Ru-VTO has the highest capacity at 3% ruthenium content and excellent cycling stability (Figure S9). Therefore, Ru doping was particularly successful in enhancing the electrochemical characteristics of VTO nanoparticles. Figure 3d shows that the current density gradually increases with the charge/discharge curves at rate stages 50, 100, 200, 500, 1000, 2000 and 5000 mA g$^{-1}$. The charge/discharge curve of Ru-VTO remains stable, with the charge capacity decreasing from 297 mAh g$^{-1}$ to 237, 194, 166, 138 and 83 mAh g$^{-1}$, respectively. Additionally, the capacity returned to 240 mAh g$^{-1}$ after a protracted period of high-speed cycling when the current density was turned to 50 mA g$^{-1}$, which unequivocally proved that the Ru-VTO sample had an exceptional rate capacity. Ru-VTO was further subjected to cycling at a current density of 1000 mA g$^{-1}$ (Figure 3e), whose capacity remained at 104 mAh g$^{-1}$ even after 2000 cycles. Moreover, the Coulomb efficiency was close to 100% in each cycle.

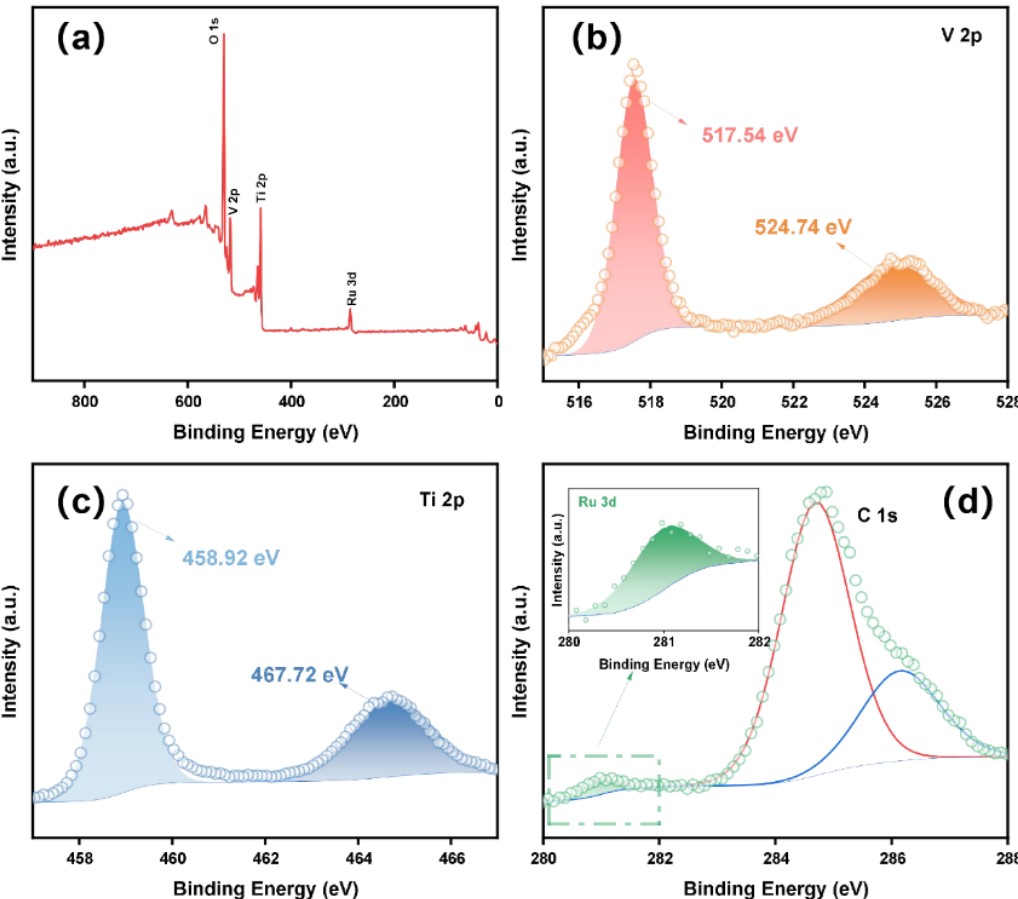

**Figure 2.** (**a**) The XPS survey spectra of Ru-VTO. (**b–d**) XPS spectra of V 2p, Ti 2p and Ru 3d in the initial state.

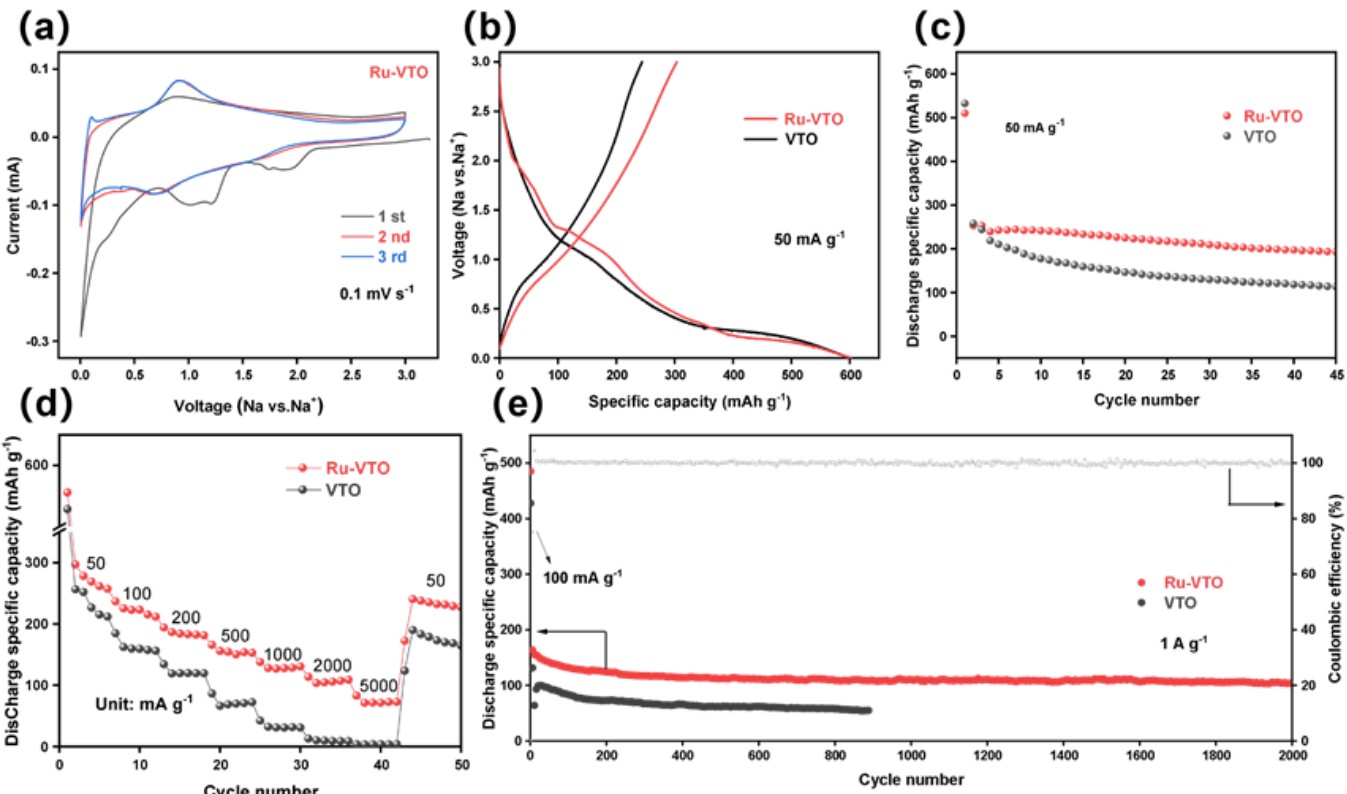

**Figure 3.** (**a**) CV test result for the initial three cycles of Ru-VTO at 0.1 mV s$^{-1}$. (**b**) Initial galvanostatic discharge$-$charge voltage profiles at 50 mA g$^{-1}$, (**c**) cycling performance at 50 mA g$^{-1}$, (**d**) rate performance at a different density, and (**e**) long-term cycling stability at 1000 mA g$^{-1}$ for Ru-VTO and VTO.

Based on the equation $i = av^b$ [47], the surface type (capacitive contribution) of the electrochemical behavior ($i$) is estimated with the current ($v$) as a function of the scan rate. The log($i$)-log($v$) curve can be utilized to determine the value of b. In more specific terms, b = 1 implies that the storage of Na$^+$ is totally dominated during the capacitive process, whereas b = 0.5 suggests that the storage of Na$^+$ is completely dominated during the diffusive process.

As shown in Figures 4a and S3b, the anode peak shift of Ru-VTO is much smaller than that of the original VTO as the scan rate increases, demonstrating a notable improvement in the Na ion diffusion. The higher b value of Ru-VTO (0.89/0.83) is determined by the cathode/anode, indicating a more favorable surface capacitance domination of Ru-VTO (Figure 4b). In addition, the behavior of the capacitive element was examined by Dunn's approach. Diffusion-dominated charges and capacitive-dominated charges can be separated from the overall charge storage volume at a certain rate (0.1–1 mV s$^{-1}$) in accordance with the equation $i(v) = k_1 v + k_2 v^{\frac{1}{2}}$ [48,49]. With a capacitive contribution of 88.4% at 0.8 mV s$^{-1}$ (shaded region), Ru-VTO outperformed VTO, which contributed 80.1% of the overall capacity (Figure S3a), explaining the fundamental reason for the outstanding rate performance of Ru-VTO (Figure 4c). The capacitive contribution of Ru-VTO became increasingly clear as the scan rate rose, peaking at 93% when the scan rate was increased to 1 mV s$^{-1}$, as seen in Figure 4d–e. Such an excellent capacitive performance of Ru-VTO was mainly related to the improved conductivity after Ru doping. It can enhance the storage of the surface charge and effectively improve the fast pseudo-capacitance process at the interface between Ru-VTO and electrolyte. The electrochemical impedance spectra (EIS) curves of Ru-VTO and VTO are shown in Figure S7; Ru-VTO electrodes have a lower charge transfer resistance (845Ω) than VTO electrodes (1488Ω). The Rct and charge transfer

resistance of Ru-VTO were decreased compared with those of VTO, which was related to the augmentation in diffusion dynamics and electrical conductivity caused by Ru-doping. The EIS of the cells after the galvanostatic tests is shown in Figure S10. The diffusion kinetics in Ru-VTO and VTO were studied using a constant-flow intermittent titration technique (GITT). The Na$^+$ diffusion coefficient ($D^{GITT}$) can be calculated according to this equation: $D^{GITT} = \frac{4}{\pi\tau}\left(\frac{m_B V_M}{M_B S}\right)^2 \left(\frac{\Delta E_s}{\Delta E_\tau}\right)^2$ [50]. The potential difference during the constant current pulse is denoted as $\Delta E\tau$, and the potential difference during the open circuit is denoted as $\Delta Es$. $D^{GITT}$ (cm$^2$ s$^{-1}$) is the Na$^+$ diffusion rate, and $m_B$, $M_B$, $V_M$, $\tau$, and S are mass, molar mass, molar volume, constant current pulse time and electrode area, respectively. The results of the GITT test are shown in Figure 4f; the average diffusion rate of Na$^+$ in the Ru-VTO electrode during discharge is $5.9 \times 10^{-9}$ cm$^2$ s$^{-1}$, while VTO is only $1.7 \times 10^{-9}$ cm$^2$ s$^{-1}$ and Ru-VTO delivers a theoretical discharge capacity of 343 mAh g$^{-1}$, while VTO is only 256 mAh g$^{-1}$ (Figure S4a,b). Ru-VTO demonstrated a higher diffusion coefficient than VTO, further demonstrating the enhanced electrochemical reaction kinetics brought on by Ru-doping.

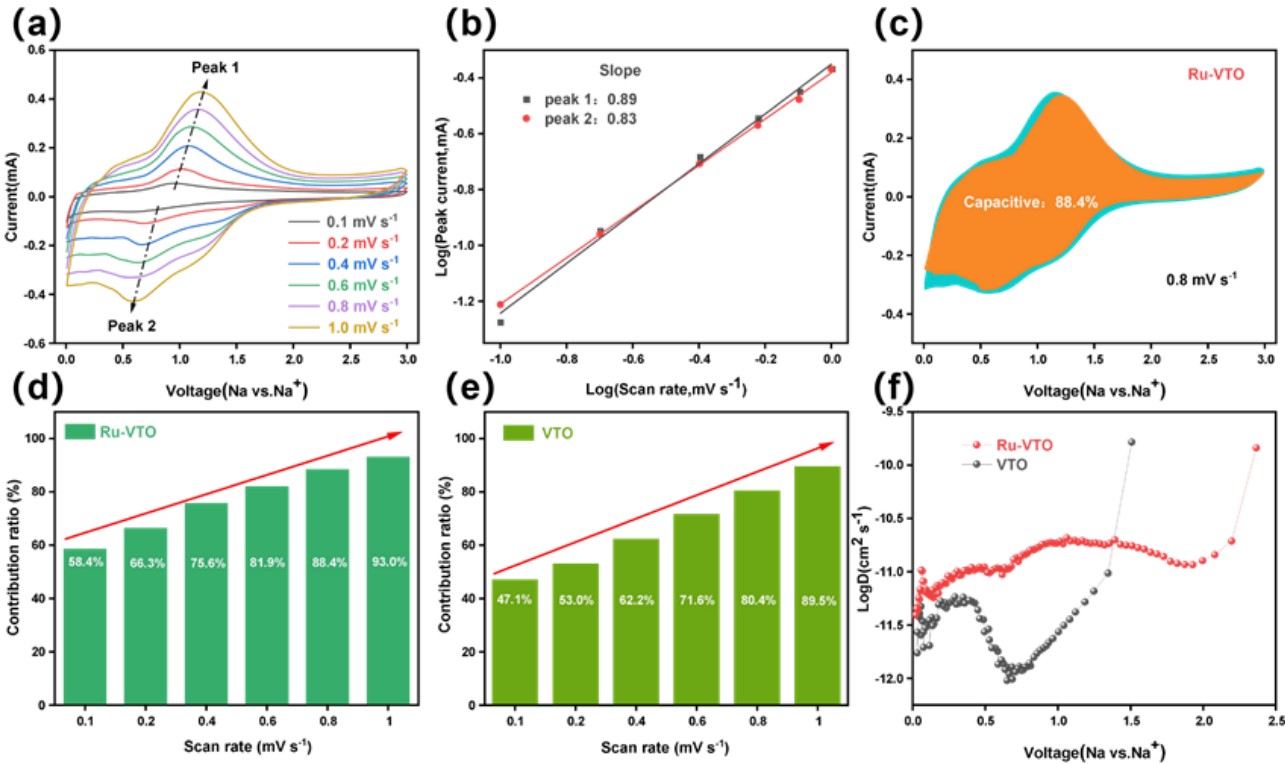

**Figure 4.** (**a**) CV curves of Ru-VTO at scan rates range from 0.1 to 1.0 mV s$^{-1}$. (**b**) Linear relationship of log($i$) vs. log($v$). (**c**) The contribution ratio of the capacitive capacities at 0.8 mV s$^{-1}$. (**d**,**e**) The contribution ratio of the capacitive capacities at different scan rates for Ru-VTO and VTO. (**f**) Evolution of the diffusion coefficient during the discharging process.

## 4. Conclusions

In summary, based on a straightforward hydrothermal synthesis procedure and subsequent heat treatments, a ruthenium-doped anatase-type VTi$_{2.6}$O$_{7.2}$ ultrafine nanocrystal was synthesized. The electronic conductivity of VTO can be enhance through Ru-doping, which greatly facilitates the electronic/ionic transport mechanism during the sodium storage process. As an anode material for sodium-ion batteries, Ru-VTO provides a reversible capacity of up to 297 mAh g$^{-1}$ at 50 mA g$^{-1}$ and maintains a capacity of 104 mAh g$^{-1}$ when cycled at a high current density of 1000 mA g$^{-1}$ for 2000 cycles, exhibiting an excellent rate cycling performance compared with VTO (51 mAh g$^{-1}$ after 900 cycles). The



great potential of Ru-VTO electrodes for SIB applications is described in this study, which provides directions for the development of new excellent-performance electrode materials.

**Supplementary Materials:** The following supporting information can be downloaded at: https://www.mdpi.com/article/10.3390/coatings13030490/s1, Figure S1: SEM images of (a) Ru-VTO and (b) VTO; Figure S2: XRD pattern of Ru-VTO precursor; Figure S3: (a) The contribution ratio of the capacitive capacities at 0.8 mV s$^{-1}$ for VTO. (b) CV curves of VTO at scan rates range from 0.1 to 1.0 mV s$^{-1}$; Figure S4: Potential response during GITT measurements at 20 mA g$^{-1}$ for (a) Ru-VTO and (b) VTO; Figure S5: Galvanostatic charge and discharge voltage profiles of (a) Ru-VTO and (b) VTO under different current densities; Figure S6: CV test result for the initial three cycles of Ru-VTO at 0.1 mV s$^{-1}$; Figure S7: EIS curves of Ru-VTO and VTO; Figure S8: Cycling performance at 50 mA g$^{-1}$ for TiO$_2$, VTO and Ru-VTO; Figure S9: Cycling performance of Ru-doped samples with different concentrations at 50 mA g$^{-1}$; Figure S10: EIS of Ru-VTO after the galvanostatic tests; Figure S11: XPS spectra of O 1s for TiO$_2$, VTO and Ru-VTO; Figure S12: Element mapping of Ru-VTO; Figure S13: The quantitative EDS analysis for Ru-VTO; Equation (S1): Scherrer equation; Table S1: Elemental content analysis of XPS for Ru-VTO.

**Author Contributions:** Conceptualization, methodology, X.W., C.H., L.Z. and K.H.; data curation, writing—original draft preparation, writing—review and editing, G.Z., J.L., J.C. and H.W.; project administration, funding acquisition, X.W. All authors have read and agreed to the published version of the manuscript.

**Funding:** This work was supported by the Hainan Provincial Joint Project of Sanya Yazhou Bay Science and Technology City (520LH055, 2021JJLH0069, 2021CXLH0007) and the Sanya Science and Education Innovation Park of Wuhan University of Technology (2021KF0016, 2021KF0019, 2021KF0020).

**Institutional Review Board Statement:** Not applicable.

**Informed Consent Statement:** Not applicable.

**Data Availability Statement:** Data sharing is not applicable to this work.

**Conflicts of Interest:** The authors declare no conflict of interest.

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
