# Peer review of "A Ru-Doped VTi2.6O7.2 Anode with High Conductivity for Enhanced Sodium Storage"

_coatings, doi:10.3390/coatings13030490_

Round 1

Reviewer 1 Report

The manuscript entitled: “Ru-doped VTi2.6O7.2 anode with high conductivity for enhanced sodium storage” submitted to Coatings -MDPI as Article describes hydrothermal preparation and characterization of titania based anodic material for sodium-ion batteries. Authors characterised the electrode material using XRD, XPS, SEM-EDS, galvanostatic charge-discharge studies, cyclic voltammetry and electrochemical impedance spectroscopy. The obtained results confirmed the supremacy of Ru-doped material over the pristine in terms of higher available capacity and long-term cycling stability up to 200 cycles at 100 mA/g rate.

Materials for electrode construction in different kind of batteries, especially those that could replace lithium technology are of high interest. In my opinion this manuscript is suitable to be published in current form after minor English spellcheck.

Here, I present the list of issues/questions that requires authors attention:

1.     Whole article – a thorough spellcheck should be performed to eliminate any miss spelling, i.e. in title “VT2.6O7.2”.

2.     general question: it would be interesting if authors could present results for different ratio of Ru-doping. In the article authors presented just one, the best results.

3.     General question: Could authors explain how they were able to select the most promising material? Where there any research survey or experience?

Author Response

Response Letter

Manuscript ID: coatings-2193026

Ru-doped VTi2.6O7.2 anode with high conductivity for enhanced sodium storage

Guangwan Zhang1,2, Chunhua Han1,2*, Kang Han1, Jinshuai Liu1,2, Jinghui Chen1,2, Haokai Wang1,2, Lei Zhang, 2,3* Xuanpeng Wang2,3*

Response to Reviewer-#1:

We thank the Reviewer-#1 for the thoughtful and encouraging comments about our manuscript and welcome the opportunity to address and clarify the issues raised in the report. Our responses to the points raised in the report are described below following specific Reviewer comments.

General Comment:

This study presents the electrochemical performance of ruthenium-doped anatase nanocrystal . The authors tried to evaluate the performance of this Ru-VTO as an anode for Na-ion batteries. Although, this paper needs major revision before accepting in the Coatings journal. First of all the authors should clarify the novelty and the importance of their study and highlight it in the manuscript. Besides, there are other concerns that should be addressed by the authors.

Response to General Comment:

We sincerely thank the Reviewer-#1's high remarks and important support of our work. We have carefully revised our manuscript according to the Reviewer-#1's suggestions and comments.

Comment 1-1:

Whole article–a thorough spellcheck should be performed to eliminate any miss spelling, i.e. in title “VT2.6O7.2”.

Response to Comment 1-1:

Thank you very much for Reviewer-#1's comment.

We have carefully and diligently checked the text to eliminate all spelling errors, and the revised manuscript have been re-uploaded.

Comment 1-2:

It would be interesting if authors could present results for different ratio of Ru-doping. In the article authors presented just one, the best results.

Response to Comment 1-2:

Thank you very much for Reviewer-#1's kind comments.

We have demonstrated that by adding ruthenium to the VTi2.6O7.2 sample, which is a highly electrochemically active element, the sample's ionic/electronic conductivity can be successfully improved.

Based on the above idea, we introduced three different doping concentrations of Ru into the VTO samples. The three different doping concentrations were 1%, 3% and 5% relative to the VTO sample. The electrochemical tests were performed on the 3 samples and the test results are shown below (Fig. R1). It can be found that the Ru-VTO has the highest capacity at 3% ruthenium content and excellent cycling stability.

Comment 1-3:

Could authors explain how they were able to select the most promising material? Where there any research survey or experience?

Response to Comment 1-3:

Thank you very much for Reviewer-#1's comments.

As the response to comment 1-2, With extensive experiments and tests, we find that the Ru-VTO will perform satisfactorily when the Ru doping concentration is 3%. Therefore, we believe that the Ru-VTO sample shown in the text will be the most promising material.

Overall, we sincerely thank Reviewer-#1's kind and significant comments on our manuscript. These comments have provided us an opportunity to make further revisions and to improve this manuscript. We wish that these responses, additions and improvements will satisfy Reviewer-#1's comments. We believe that our revised manuscript has reached the standard for publication in Coatings.

Reviewer 2 Report

This study presents the electrochemical performance of ruthenium-doped anatase nanocrystal . The authors tried to evaluate the performance of this Ru-VTO as an anode for Na-ion batteries. Although, this paper needs major revision before accepting in the Coatings journal. First of all the authors should clarify the novelty and the importance of their study and highlight it in the manuscript. Besides, there are other concerns that should be addressed by the authors:

1. There are some typos and grammatical errors which should be addressed.

2. The introduction part needs to be revised. The novelty of the study should be highlighted and the reason for choosing hydrothermal method and the advantages over other methods should be clarified. 

3. For better understanding of microstructure, TEM or atleast FESEM micrographs are needed

4. What was is the optimum amount of Ru?! The authors claimed that "the

atomic content of Ru is 1.54 %", is it the optimum amount?! 

5. Please mention what was the reference electrode or the cathode?!

6. Please provide EIS of the cells after the galvanostatic tests!

7. Mainly SEI form under 1 V due to decomposition of solvents! Please reconsider this part!

8. Please provide galvanostatic discharge−charge voltage profiles at 50th cycle!

Author Response

Response Letter

Manuscript ID: coatings-2193026

Ru-doped VTi2.6O7.2 anode with high conductivity for enhanced sodium storage

Guangwan Zhang1,2, Chunhua Han1,2*, Kang Han1, Jinshuai Liu1,2, Jinghui Chen1,2, Haokai Wang1,2, Lei Zhang, 2,3* Xuanpeng Wang2,3*

Response to Reviewer-#2:

We thank the Reviewer-#2 for the thoughtful and encouraging comments about our manuscript, and welcome the opportunity to address and clarify the issues raised in the report. Our responses to the points raised in the report are described below following specific Reviewer comments.

General Comment:

This study presents the electrochemical performance of ruthenium-doped anatase nanocrystal . The authors tried to evaluate the performance of this Ru-VTO as an anode for Na-ion batteries. Although, this paper needs major revision before accepting in the Coatings journal. First of all the authors should clarify the novelty and the importance of their study and highlight it in the manuscript. Besides, there are other concerns that should be addressed by the authors.

Response to General Comment:

We sincerely thank the Reviewer-#2's high remarks and important support of our work. We have carefully revised our manuscript according to the Reviewer-#2's suggestions and comments.

Comment 2-1:

There are some typos and grammatical errors which should be addressed.

Response to Comment 2-1:

Thank you very much for Reviewer-#2's comment.

We have carefully and diligently checked the text to eliminate all spelling errors, and the revised manuscript will be re-uploaded. Some of the modifications are as follows:

  1. "These studies demonstrate the potential of Ru-VTO as an anode material for advanced SIBs.” (Abstract)
  2. "The active mass loading range of electrodes is from 1.4 to 1.6 mg cm2. A Whatman glass fiber (GF/A) was employed as the separator, meanwhile propylene carbonate, fluoroethylene carbonate (95:5) and propylene carbonate were all mixed together to form a solution containing 1M NaClO4 " (Materials and Methods)
  3. "Fig. 3d shows that the current density gradually increases with the charge/discharge curves at rate stages 50, 100, 200, 500, 1000, 2000 and 5000 mA g-1." (Result)
  4. "The great potential of Ru-VTO electrodes for SIB applications is described in this study, which provides directions for the development of new excellent-performance electrode materials." (Conclusions)

Comment 2-2:

The introduction part needs to be revised. The novelty of the study should be highlighted and the reason for choosing hydrothermal method and the advantages over other methods should be clarified.

Response to Comment 2-2:

Thank you very much for Reviewer-#2's suggestions.

We have inserted into the manuscript a description highlighting the innovative nature of the study. As follws:

“A new-type ruthenium-doped anatase type VTi2.6O7.2 ultrafine nanocrystal (Ru-VTO) anode materials was designed and prepared by a straightforward hydrothermal synthesis procedure followed by annealing, for the first time.”

In addition, a new path for the development of high-performance anode materials for sodium ion batteries is opened up by Ru-VTO, which can give great capacity (297 mAh g-1 at 50 mA g-1) together with superior high multiplicity performance and cycle stability (104 mAh g1 at 1000 mA g1 after 2000 cycles), demonstrating the applicability of our research.

Based on the previous work, we synthesized Ru-VTO samples by a straightforward hydrothermal synthesis procedure followed by annealing. Due to its benefits of homogeneous particle distribution, small particle size, and excellent purity [1], we chose the hydrothermal approach to synthesis Ru-VTO samples.

  1. Sanad MM.; Rashad MM.; Powers K., Enhancement of the electrochemical performance of hydrothermally prepared anatase nanoparticles for optimal use as high capacity anode materials in lithium ion batteries (LIBs). Applied Physics A 2015, 118:665-74.

Comment 2-3:

For better understanding of microstructure, TEM or at least FESEM micrographs are needed

Response to Comment 2-3:

Thank you very much for Reviewer-#2's suggestions. The SEM with higher separation rate is shown below (Fig. R1)

Comment 2-4:

What was is the optimum amount of Ru?! The authors claimed that "the atomic content of Ru is 1.54 %", is it the optimum amount?!

Response to Comment 2-4:

Thank you very much for Reviewer-#2's kind comments.

We have demonstrated that by adding ruthenium to the VTi2.6O7.2 sample, which is a highly electrochemically active element, the sample's ionic/electronic conductivity can be successfully improved.

Based on the above idea, we introduced three different doping concentrations of Ru into the VTO samples. The three different doping concentrations were 1%, 3% and 5% relative to the VTO sample. The electrochemical tests were performed on the 3 samples and the test results are shown below (Fig. R1). It can be found that the Ru-VTO has the highest capacity at 3% ruthenium content and excellent cycling stability. Ru is found to be 1.54% of the Ru-VTO sample when the Ru doping concentration is 3%.

In summary, we confirm that the Ru-VTO will perform satisfactorily when the Ru doping concentration is 3%.

Comment 2-5:

Please mention what was the reference electrode or the cathode?!

Response to Comment 2-5:

Thank you very much for Reviewer-#2's comments.

We have inserted a more detailed description of the cell structure in the manuscript. As follows:

“The half 2016 coin-type cells with sodium metal as the counter electrode were used to investigate the electrochemical performance.”

Comment 2-6:

Please provide EIS of the cells after the galvanostatic tests!

Response to Comment 2-6:

Thank you very much for Reviewer-#1's kind comments. The EIS of the cells after the galvanostatic tests is shown blow:

Comment 2-7:

Mainly SEI form under 1 V due to decomposition of solvents! Please reconsider this part!

Response to Comment 2-7:

Thank you very much for Reviewer-#2's kind comments.

The SEI layers and certain secondary reactions (irreversible reactions) between the electrolyte and the electrode were observed to coincide with a bright outflow plateau with a slope extending from 1.31 to 0.9 V, and this conclusion has been reported [2-6].

  1. Chen, J.; Ding, Z.; Wang, C.; Hou, H.; Zhang, Y.; Wang, C.; Zou, G.; Ji, X., Black anatase titania with ultrafast sodium-storage performances stimulated by oxygen vacancies. ACS Applied Materials & Interfaces 2016, 8 (14), 9142-9151.
  2. He, H.; Gan, Q.; Wang, H.; Xu, G.-L.; Zhang, X.; Huang, D.; Fu, F.; Tang, Y.; Amine, K.; Shao, M., Structure-dependent performance of TiO2/C as anode material for Na-ion batteries. Nano Energy 2018, 44, 217-227.
  3. McNulty, D.; Carroll, E.; O'Dwyer, C., Rutile TiO2 inverse opal anodes for Li-ion batteries with long cycle life, high-rate capability, and high structural stability. Advanced Energy Materials 2017, 7 (12) 1602291.
  4. Zhang, Y.; Ding, Z.; Foster, C. W.; Banks, C. E.; Qiu, X.; Ji, X., Oxygen vacancies evoked blue TiO2(B) nanobelts with efficiency enhancement in sodium storage behaviors. Advanced Functional Materials 2017, 27 (27), 1700856.
  5. Gan, Q.; He, H.; Zhu, Y.; Wang, Z.; Qin, N.; Gu, S.; ... & Lu, Z., Defect-assisted selective surface phosphorus doping to enhance rate capability of titanium dioxide for sodium ion batteries. ACS nano 2019, 13(8), 9247-9258.

Comment 2-8:

Please provide galvanostatic discharge−charge voltage profiles at 50th cycle!

Response to Comment 2-8:

Thank you very much for Reviewer-#2's kind comments. The galvanostatic discharge−charge voltage profiles of Ru-VTO at 50th cycle is shown blow:

Overall, we sincerely thank Reviewer-#2's kind and significant comments on our manuscript. These comments have provided us an opportunity to make further revisions and to improve this manuscript. We wish that these responses, additions and improvements will satisfy Reviewer-#2's comments. We believe that our revised manuscript has reached the standard for publication in Coatings.

Reviewer 3 Report

1. The chemical formula in the title should be corrected to be VTi2.6O7.2

2. The novelty and significance of the work must be clarified well highlighted in the introduction part.

3. The author should cite the relevant studies about doped TiO2 and various metal oxides as anodes for batteries including: Applied Physics A 118 (2015) 665-674

4. The EDS analysis for Ru-doped VTiO2 should be provided.

5. The XPS spectra for O1s must be analyzed and compared for pure TiO2, VTiO2 and Ru-doped VTiO2 to distinguish the differences in the oxygen vacancies and provide additional confirmation about the conductivity considering the following published works e.g. Materials Science in Semiconductor Processing 143 (2022) 106567

6. It is recommended to compare the obtained capacity values with previous anode of doped TiO2 whether for Li+ or Na+ ion batteries.

7. The whole paper need to be polished to improve the English language and remove the typing errors

Author Response

Response Letter

Manuscript ID: coatings-2193026

Ru-doped VTi2.6O7.2 anode with high conductivity for enhanced sodium storage

Guangwan Zhang1,2, Chunhua Han1,2*, Kang Han1, Jinshuai Liu1,2, Jinghui Chen1,2, Haokai Wang1,2, Lei Zhang, 2,3* Xuanpeng Wang2,3*

Response to Reviewer-#3:

We sincerely thank the Reviewer-#3's high remarks and important support of our work. We have carefully revised our manuscript according to the Reviewer-#3's suggestions and comments. Our responses to the points raised in the report are described below following specific Reviewer comments.

Comment 3-1:

The chemical formula in the title should be corrected to be VTi2.6O7.2.

Response to Comment 3-1:

Thank you very much for Reviewer-#3's comment.

We have carefully and diligently checked the text to eliminate all spelling errors, and the revised manuscript have been re-uploaded.

Comment 3-2:

The novelty and significance of the work must be clarified well highlighted in the introduction part.

Response to Comment 3-2:

Thank you very much for Reviewer-#3's kind comments.

We have inserted into the manuscript a description highlighting the innovative nature of the study. As follws:

“A new-type ruthenium-doped anatase type VTi2.6O7.2 ultrafine nanocrystal (Ru-VTO) anode materials was designed and prepared by a straightforward hydrothermal synthesis procedure followed by annealing, for the first time.”

In addition, a new path for the development of high-performance anode materials for sodium ion batteries is opened up by Ru-VTO, which can give great capacity (297 mAh g-1 at 50 mA g-1) together with superior high multiplicity performance and cycle stability (104 mAh g-1 at 1000 mA g-1 after 2000 cycles), demonstrating the applicability of our research.

Comment 3-3:

The author should cite the relevant studies about doped TiO2 and various metal oxides as anodes for batteries including: Applied Physics A 118 (2015) 665-674

Response to Comment 3-3:

Thank you very much for Reviewer-#3's comments.

We have cited the relevant studies about doped TiO2 and various metal oxides as anodes for batteries, and reviewer-#3-recommended articles are cited in the manuscript ([33])

  1. Sanad, M.; Rashad, MM.; Powers, K., Enhancement of the electrochemical performance of hydrothermally prepared anatase nanoparticles for optimal use as high capacity anode materials in lithium ion batteries (LIBs). Applied Physics A 2015, 118:665-74.

Comment 3-4:

The EDS analysis for Ru-doped VTiO2 should be provided.

Response to Comment 3-4:

Thank you very much for Reviewer-#3's comments.

We have inserted the relevant description in the manuscript. As follows:

“The elemental distribution of Ru-VTO was further observed by EDS-mapping, and the results revealed that V, Ti, O, and Ru were consistently distributed in the sample (Fig. 1d), indicating that Ru is successfully doped into the VTO sample.”

Comment 3-5:

The XPS spectra for O1s must be analyzed and compared for pure TiO2, VTiO2 and Ru-doped VTiO2 to distinguish the differences in the oxygen vacancies and provide additional confirmation about the conductivity considering the following published works e.g. Materials Science in Semiconductor Processing 143 (2022) 106567

Response to Comment 3-5:

Thank you very much for Reviewer-#3's comments. XPS spectra of O 1s for TiO2, VTO and Ru-VTO is shown in Fig. R1

According to Fig. R1, Ru-VTO exhibits an oxygen vacancy signal (O-Vac.). Meanwhile, the O-peak location of Ru-VTO moves to a lower energy state. The results mentioned above also show that Ru doping to into the VTO sample resulted in the generation of more oxygen vacancies, which explains that Ru-VTO has superior ion/electron conductivity [1].

  1. Sanad MM.; Azab AA,; Taha TA., Introduced oxygen vacancies in cadmium ferrite anode materials via Zn2+ incorporation for high performance lithium-ion batteries. Materials Science in Semiconductor Processing 2022, 1; 143:106567.

Comment 3-6:

It is recommended to compare the obtained capacity values with previous anode of doped TiO2 whether for Li+ or Na+ ion batteries.

Response to Comment 3-6:

Thank you very much for Reviewer-#3's suggestions. cycling performance at 50 mA g-1 for TiO2 VTO and Ru-VTO is shown in Fig. R2

As demonstrated in Fig. R2, compared to pure TiO2, VTO has a higher capacity and a superior energy storage potential, but with subsequent charge/discharge experiments, the VTO solid solution capacity decreases significantly. The above results indicate that the VTO solid solution offers higher potential for energy storage, but its cycle stability has declined. Ru was added to the VTO solid solution as a dopant, and Ru-VTO demonstrated more capacity than pure TiO2 while also maintaining superior cycling stability than the VTO solid solution.

Comment 3-7:

The whole paper need to be polished to improve the English language and remove the typing errors

Response to Comment 3-7:

Thank you very much for Reviewer-#3's suggestions.

We have carefully and diligently checked the text to eliminate all spelling errors, and the manuscript has been touched up. The new manuscript have been re-uploaded. Some of the modifications are as follows:

  1. "These studies demonstrate the potential of Ru-VTO as an anode material for advanced SIBs.” (Abstract)
  2. "The active mass loading range of electrodes is from 1.4 to 1.6 mg cm2. A Whatman glass fiber (GF/A) was employed as the separator, meanwhile propylene carbonate, fluoroethylene carbonate (95:5) and propylene carbonate were all mixed together to form a solution containing 1M NaClO4 " (Materials and Methods)
  3. "Fig. 3d shows that the current density gradually increases with the charge/discharge curves at rate stages 50, 100, 200, 500, 1000, 2000 and 5000 mA g-1." (Result)
  4. "The great potential of Ru-VTO electrodes for SIB applications is described in this study, which provides directions for the development of new excellent-performance electrode materials." (Conclusions)

Overall, we sincerely thank Reviewer-#3's kind and significant comments on our manuscript. These comments have provided us an opportunity to make further revisions and to improve this manuscript. We wish that these responses, additions and improvements will satisfy Reviewer-#3's comments. We believe that our revised manuscript has reached the standard for publication in Coatings.

Reviewer 4 Report

Comments to the Author

This article demonstrates using ruthenium-doped anatase-type ultrafine nanocrystal VTi2.6O7.2 as an anode material for Sodium-ion batteries. The formation of the solid solution via vanadium, followed by doping Ru, is investigated here, a new anode material for sodium-ion batteries. The electrochemical results are interesting, with cyclability up to 2000 cycles and better rate performance. However, there are a few suggestions to improve the manuscript;

1.      The importance of a solid solution should be highlighted in the introduction section. What is the theoretical capacity of VTi2.6O7.2? Does the theoretical capacity be affected by the formation of the solid solution?

2.      Electrochemical studies for pristine anatase TiO2 can be compared in the supplementary section to better understand the solid solution's influence.

3.      In Fig 3 (a), the authors are suggested to explain the peak at 0.1 V in the third cycle.

4.      Authors should justify using Ru as a doping agent over other metal ions.

5.      It is suggested to recheck the XPS spectra fitting for V and Ti. It is difficult to observe the fitting for Ru due to the relative intensity with the C 1s peak. It is recommended to show a magnified image as an inset for Ru.

6.      The diffusion coefficient values are missing in the manuscript.

7.      The Nyquist plot should be fitted with an equivalent circuit. The Rct values are missing in the manuscript.

8.    The manuscript would benefit from careful re-reading and language polishing before submitting the revised version. 

Author Response

Response Letter

Manuscript ID: coatings-2193026

Ru-doped VTi2.6O7.2 anode with high conductivity for enhanced sodium storage

Guangwan Zhang1,2, Chunhua Han1,2*, Kang Han1, Jinshuai Liu1,2, Jinghui Chen1,2, Haokai Wang1,2, Lei Zhang, 2,3* Xuanpeng Wang2,3*

Response to Reviewer-#4:

We thank the Reviewer-#4 for the thoughtful and encouraging comments about our manuscript, and welcome the opportunity to address and clarify the issues raised in the report. Our responses to the points raised in the report are described below following specific Reviewer comments.

General Comment:

This article demonstrates using ruthenium-doped anatase-type ultrafine nanocrystal VTi2.6O7.2 as an anode material for Sodium-ion batteries. The formation of the solid solution via vanadium, followed by doping Ru, is investigated here, a new anode material for sodium-ion batteries. The electrochemical results are interesting, with cyclability up to 2000 cycles and better rate performance. However, there are a few suggestions to improve the manuscript;

Response to General Comment:

We sincerely thank the Reviewer-#4's high remarks and important support of our work. We have carefully revised our manuscript according to the Reviewer-#4's suggestions and comments.

Comment 4-1:

The importance of a solid solution should be highlighted in the introduction section. What is the theoretical capacity of VTi2.6O7.2? Does the theoretical capacity be affected by the formation of the solid solution?

Response to Comment 4-1:

Thank you very much for Reviewer-#4's comments.

We have inserted a description of the importance of solid solution in the manuscript. As follows:

“The incorporation of the guest element into the matrix to form a solid solution can effectively lower the energy potential barrier for sodium ion diffusion, increasing conductivity. In addition, the composition of solid solutions can be more easily adjusted, and their easy adjustment properties further lead to a diversity of active materials. Previous studies have demonstrated that by changing interatomic interactions, the creation of substituted solid solutions can significantly improve electrochemical performance [26, 27].”

The capacity of the VTO electrode is derived from the conversion of V4+ and Ti4+ to V3+ and Ti3+. Therefore, 1 mol VTi2.6O7.2 will provide 3.6 mol electron transfer number, which means the theoretical capacity of VTO is 368.5 mAh g-1.

As shown in Fig. R1, the capacity of pure TiO2 is lower than that of the VTO sample, indicating that the development of solid solution is the cause of the capacity increase.

Comment 4-2:

Electrochemical studies for pristine anatase TiO2 can be compared in the supplementary section to better understand the solid solution's influence.

Response to Comment 4-2:

Thank you very much for Reviewer-#4's suggestions. Cycling performance at 50 mA g-1 for TiO2, VTO and Ru-VTO is shown in Fig. R1

As demonstrated in Fig. R1, compared to pure TiO2, VTO has a higher capacity and a superior energy storage potential, but with subsequent charge/discharge experiments, the VTO solid solution capacity decreases significantly. The above results indicate that the VTO solid solution offers higher potential for energy storage, it can be concluded that the higher capacity exhibited by VTO is attributed to the formation of solid solution. However, the cycle stability of VTO has declined. Therefore, Ru was chosen as a dopant to produce Ru-VTO to enhance the conductivity and cycling stability of VTO samples. Ru-VTO demonstrated more capacity than pure TiO2 while also maintaining superior cycling stability than the VTO solid solution. We have added the Figure R1 to the support information (Fig. S8).

Comment 4-3:

In Fig 3 (a), the authors are suggested to explain the peak at 0.1 V in the third cycle.

Response to Comment 4-3:

Thank you very much for Reviewer-#4's comments.

The third cycle of CV curve has been displayed separately, with the peculiar portion highlighted, to enable a clearer and more intuitive inspection. As shown in Fig. R2, a peak appears at 0.1 V during charging and a broad peak at 1.57 V during subsequent discharging, and it should be noted that this peak appears only during the third cyclic voltammetry testing. The result is attributed to the change of V4+and Ti4+ into V3+and Ti3+, respectively [1].

  1. Sheng, J.; Peng, C.; Yan, S.; Zhang, G.; Jiang, Y.; An, Q.; Wei, Q.; Ru, Q.; Mai, L., New anatase phase VTi6O7.2 ultrafine nanocrystals for high-performance rechargeable magnesium-based batteries. Journal of Materials Chemistry A 2018, 6 (28), 13901-13907.

Comment 4-4:

Authors should justify using Ru as a doping agent over other metal ions.

Response to Comment 4-4:

Thank you very much for Reviewer-#4's comments.

Firstly, Ruthenium effectively enhanced the electrical conductivity of VTO as Ru is one of the few elements with excellent electrochemical activity [2]. We have demonstrated this phenomenon through experiments and tests.

Secondly, the application of Ru as a dopant to enhance the conductivity of titanium-based anode materials can be considered novel, which was not considered by previous researchers.

Finally, from what has been mentioned above, we decided to modify the VTO solid solution by adding ruthenium as a dopant.

  1. Xu, Z.; Yin, X.; Guo, Y.; Pu, Y.; & He, M.; Ru-Doping in TiO2 electron transport layers of planar heterojunction perovskite solar cells for enhanced performance. Journal of Materials Chemistry C 2018, 6(17), 4746-4752.

Comment 4-5:

It is suggested to recheck the XPS spectra fitting for V and Ti. It is difficult to observe the fitting for Ru due to the relative intensity with the C 1s peak. It is recommended to show a magnified image as an inset for Ru.

Response to Comment 4-5:

Thank you very much for Reviewer-#4's suggestions.

After our careful verification, the XPS spectra fits of Ti and V are correct and can be found in the published works: J. Mater. Chem. A, 2018, 6, 13901-13907.

Magnified image as an inset for Ru is shown in Fig. R3,and we have added the Figure R3 to the manuscript.

Comment 4-6:

The diffusion coefficient values are missing in the manuscript.

Response to Comment 4-6:

Thank you very much for Reviewer-#4's comments.

We have inserted a description of the diffusion coefficient in the manuscript. As follows:

“The average diffusion rate of Na+ in the Ru-VTO electrode during discharge is 5.9×10-9 cm2 s-1, while VTO is only 1.7×10-9 cm2 s-1.”

Comment 4-7:

The Nyquist plot should be fitted with an equivalent circuit. The Rct values are missing in the manuscript.

Response to Comment 4-7:

Thank you very much for Reviewer-#4's suggestions.

The equivalent circuit has been installed in the Nyquist plot (Fig. R4), which has been updated in the support information

We have inserted a description of the Rct value in the manuscript. As follows:

“Ru-VTO electrodes have a lower charge transfer resistance (845Ω) than VTO electrodes (1488Ω).”

Comment 4-8:

The manuscript would benefit from careful re-reading and language polishing before submitting the revised version.

Response to Comment 4-8:

Thank you very much for Reviewer-#4's kind comments.

We have carefully checked and improved the English writing in the revised manuscript, and the revised manuscript have been re-uploaded. Some of the modifications are as follows:

  1. "These studies demonstrate the potential of Ru-VTO as an anode material for advanced SIBs.” (Abstract)
  2. "The active mass loading range of electrodes is from 1.4 to 1.6 mg cm2. A Whatman glass fiber (GF/A) was employed as the separator, meanwhile propylene carbonate, fluoroethylene carbonate (95:5) and propylene carbonate were all mixed together to form a solution containing 1M NaClO4 " (Materials and Methods)
  3. "Fig. 3d shows that the current density gradually increases with the charge/discharge curves at rate stages 50, 100, 200, 500, 1000, 2000 and 5000 mA g-1." (Result)
  4. "The great potential of Ru-VTO electrodes for SIB applications is described in this study, which provides directions for the development of new excellent-performance electrode materials." (Conclusions)

Overall, we sincerely thank Reviewer-#4's kind and significant comments on our manuscript. These comments have provided us an opportunity to make further revisions and to improve this manuscript. We wish that these responses, additions and improvements will satisfy Reviewer-#4's comments. We believe that our revised manuscript has reached the standard for publication in Coatings.

Reviewer 5 Report

Guangwan Zhang and co-workers report a ruthenium-doped anatase type VTi2.6O7.2 anode with high conductivity for enhanced sodium storage. They report improvement of performance of VTi2.6O7.2 as anode by Ru doping. The results are clear and may attract readets. However, Some point are confusing and should be revised. My major concerns are follows:

1.       The chemical formula in the title is wrong. VT2.6O7.2 would be VTi2.6O7.2.

2.       Cell structure is not clear. What was a reference electrode or a counter electrode?

3.       What was the evidence for ultrafine nanocrystal? How small is the crystallite size estimated by applying the t Scherrer equation to XRD profile in Figure 1(b).

4.       The description of “XRD spectra” in page 3 is wrong, because it is not energy depence. It should be XRD profiles or XRD patterns.

5.       Description of “, τ, and S are mass, molar mass, molar volume, constant current pulse time, and electrode area, respectively.” is an incorrect sentence, because number of symbols is only two, while that of explanation is five.

6.       In addition to the item 4 and 5, descriptions and English should be improved with native speakers.

From above reasons, the authors are advised to revise the manuscript.

Author Response

Response Letter

Manuscript ID: coatings-2193026

Ru-doped VTi2.6O7.2 anode with high conductivity for enhanced sodium storage

Guangwan Zhang1,2, Chunhua Han1,2*, Kang Han1, Jinshuai Liu1,2, Jinghui Chen1,2, Haokai Wang1,2, Lei Zhang, 2,3* Xuanpeng Wang2,3*

Response to Reviewer-#5:

We thank the Reviewer-#5 for the thoughtful and encouraging comments about our manuscript, and welcome the opportunity to address and clarify the issues raised in the report. Our responses to the points raised in the report are described below following specific Reviewer comments.

General Comment:

Guangwan Zhang and co-workers report a ruthenium-doped anatase type VTi2.6O7.2 anode with high conductivity for enhanced sodium storage. They report improvement of performance of VTi2.6O7.2 as anode by Ru doping. The results are clear and may attract readets. However, Some point are confusing and should be revised.

Response to General Comment:

We sincerely thank the Reviewer-#5's high remarks and important support of our work. We have carefully revised our manuscript according to the Reviewer-#5's suggestions and comments.

Comment 5-1:

The chemical formula in the title is wrong. VT2.6O7.2 would be VTi2.6O7.2.

Response to Comment 5-1:

Thank you very much for Reviewer-#5's kind comments.

We have carefully and diligently checked the text to eliminate all spelling errors, and the revised manuscript have been re-uploaded.

Comment 5-2:

Cell structure is not clear. What was a reference electrode or a counter electrode?

Response to Comment 5-2:

Thank you very much for Reviewer-#5's comments.

We have inserted a more detailed description of the cell structure in the manuscript. As follows:

“The half 2016 coin-type cells with sodium metal as the counter electrode were used to investigate the electrochemical performance. ”

Comment 5-3:

What was the evidence for ultrafine nanocrystal? How small is the crystallite size estimated by applying the Scherrer equation to XRD profile in Figure 1(b).

Response to Comment 5-3:

Thank you very much for Reviewer-#5's comments.

Scherrer Equation:  , is the grain size, , , , and  are Scherrer constant, X-ray wavelength, Diffraction peak half-height width, and Cape Prague respectively. The value of  is 0.89,and  is 0.154056 nm, while  and  need to be obtained by XRD profile.

We calculated the grain size of Ru-VTO to be 10.66 nm by applying the Scherrer equation to the XRD profile.

Consequently, we believe that the Ru-VTO samples are made of ultrafine nanocrystals.

Comment 5-4:

The description of “XRD spectra” in page 3 is wrong, because it is not energy depence. It should be XRD profiles or XRD patterns.

Response to Comment 5-4:

Thank you very much for Reviewer-#5's kind comments.

We have changed "XRD spectra" to "XRD profiles" in the manuscript, and the revised manuscript have been uploaded.

Comment 5-5:

Description of “τ, and S are mass, molar mass, molar volume, constant current pulse time, and electrode area, respectively.” is an incorrect sentence, because number of symbols is only two, while that of explanation is five.

Response to Comment 5-5:

Thank you very much for Reviewer-#5's comments.

We have revised the manuscript and a new manuscript have been uploaded. The specific replacement part was as follows:

“ (cm2 s-1) is the Na+ diffusion rate, and , , , τ, and S are mass, molar mass, molar volume, constant current pulse time, and electrode area, respectively.”

Comment 5-6:

In addition to the item 4 and 5, descriptions and English should be improved with native speakers.

Response to Comment 5-6:

Thank you very much for Reviewer-#5's kind comments.

We have carefully checked and improved the English writing in the revised manuscript, and the revised manuscript have been re-uploaded.

1."These studies demonstrate the potential of Ru-VTO as an anode material for advanced SIBs.” (Abstract)

  1. "The active mass loading range of electrodes is from 1.4 to 1.6 mg cm2. A Whatman glass fiber (GF/A) was employed as the separator, meanwhile propylene carbonate, fluoroethylene carbonate (95:5) and propylene carbonate were all mixed together to form a solution containing 1M NaClO4 " (Materials and Methods)
  2. "Fig. 3d shows that the current density gradually increases with the charge/discharge curves at rate stages 50, 100, 200, 500, 1000, 2000 and 5000 mA g-1." (Result)
  3. "The great potential of Ru-VTO electrodes for SIB applications is described in this study, which provides directions for the development of new excellent-performance electrode materials." (Conclusions)

Overall, we sincerely thank Reviewer-#5's kind and significant comments on our manuscript. These comments have provided us an opportunity to make further revisions and to improve this manuscript. We wish that these responses, additions and improvements will satisfy Reviewer-#5's comments. We believe that our revised manuscript has reached the standard for publication in Coatings.

Round 2

Reviewer 2 Report

The authors have addrressed almost all of the concerns and the paper can ce accepted after a minor revision. 

For the comment 4 please provide cyclic performance in supplemantary and for comment 6 please provide the EIS data in the supplemantary manuscript 

Author Response

Response Letter

Manuscript ID: coatings-2193026

Ru-doped VTi2.6O7.2 anode with high conductivity for enhanced sodium storage

Guangwan Zhang1,2, Chunhua Han1,2*, Kang Han1, Jinshuai Liu1,2, Jinghui Chen1,2, Haokai Wang1,2, Lei Zhang, 2,3* Xuanpeng Wang2,3*

Response to Reviewer-#2:

We thank the Reviewer-#2 for the thoughtful and encouraging comments about our manuscript, and welcome the opportunity to address and clarify the issues raised in the report. Our responses to the points raised in the report are described below following specific Reviewer comments.

General Comment:

The authors have addrressed almost all of the concerns and the paper can ceaccepted after a minor revision.

Response to General Comment:

We sincerely thank the Reviewer-#2's high remarks and important support of our work. We have carefully revised our manuscript point-by-point according to the Reviewer-#2's suggestions and comments.

Comment :

For the comment 4 please provide cyclic performance in supplemantary and for comment 6 please provide the EIS data in the supplemantary manuscript

Response to Comment

Thank you very much for Reviewer-#2's comment. 

We have added the Figure R2 and Figure R3 in the revised Supplementary Materials (Fig. S9 and Fig. S10).

Overall, we sincerely thank Reviewer-#2's kind and significant comments on our manuscript. These comments have provided us an opportunity to make further revisions and to improve this manuscript. We wish that these responses, additions and improvements will satisfy Reviewer-#2's comments. We believe that our revised manuscript has reached the standard for publication in Coatings.

Reviewer 3 Report

1- The authors did not answer the comment no. 3 thus, the corresponding quantitative EDS analysis for Ru-doped VTiO2 should be provided.

2- The response for comment no. 3  must be added to the revised version of manuscript including Figs R1 and R2 and the suggested reference

3- the comparison in comment no. 6 did not provided in the revised version

Author Response

Response Letter

Manuscript ID: coatings-2193026

Ru-doped VTi2.6O7.2 anode with high conductivity for enhanced sodium storage

Guangwan Zhang1,2, Chunhua Han1,2*, Kang Han1, Jinshuai Liu1,2, Jinghui Chen1,2, Haokai Wang1,2, Lei Zhang, 2,3* Xuanpeng Wang2,3*

Response to Reviewer-#3:

We sincerely thank the Reviewer-#3's high remarks and important support of our work. We have carefully revised our manuscript according to the Reviewer-#3's suggestions and comments. Our responses to the points raised in the report are described below following specific Reviewer comments.

Comment 3-1:

The authors did not answer the comment no. 3 thus, the corresponding quantitative EDS analysis for Ru-doped VTiO2 should be provided.

Response to Comment 3-1:

Thank you very much for Reviewer-#3's comments.

Element mapping and the quantitative EDS analysis for Ru-VTO is shown in Fig. R3 and R4:

We have added the Figure R3 and R4 to the support information (Fig. S12 and S13)

Comment 3-2:

The response for comment no. 3 must be added to the revised version of manuscript including Figs R1 and R2 and the suggested reference.

Response to Comment 3-2:

Thank you very much for Reviewer-#3's comments.

We have cited the relevant studies about doped TiO2 and various metal oxides as anodes for batteries, and Reviewer-#3-recommended articles are cited in the manuscript .

  1. Sanad, M.; Rashad, MM.; Powers, K., Enhancement of the electrochemical     performance of hydrothermally prepared anatase nanoparticles for optimal use as high capacity anode materials in lithium ion batteries (LIBs). Applied Physics A 2015, 118:665-74.

We have inserted a more detailed description of the suggested reference in the manuscript. As follows:

“For example, M. M. S. Sanad reported the mesoporous TiO2 introduced by noble metal nanoparticles with enhanced electrical conductivity [33].”

We have added the Figure R1 and Figure R2 to the support information (Fig. S11 and Fig. S8)

Comment 3-3:

the comparison in comment no. 6 did not provided in the revised version.

Response to Comment 3-3:

Thank you very much for Reviewer-#3's comments.

Cycling performance at 50 mA g-1 for TiO2, VTO and Ru-VTO is shown in Fig. R2, and we have added the Figure R2 to the support information ( Fig. S8)

Overall, we sincerely thank Reviewer-#3's kind and significant comments on our manuscript. These comments have provided us an opportunity to make further revisions and to improve this manuscript. We wish that these responses, additions and improvements will satisfy Reviewer-#3's comments. We believe that our revised manuscript has reached the standard for publication in Coatings.

Reviewer 4 Report

The authors have addressed my comments, the manuscript can be considered for publication. 

Author Response

Response Letter

Manuscript ID: coatings-2193026

Ru-doped VTi2.6O7.2 anode with high conductivity for enhanced sodium storage

Guangwan Zhang1,2, Chunhua Han1,2*, Kang Han1, Jinshuai Liu1,2, Jinghui Chen1,2, Haokai Wang1,2, Lei Zhang, 2,3* Xuanpeng Wang2,3*

Response to Reviewer-#4:

We thank very much for the Reviewer-#4s' valuable comments and suggestions. These comments have provided us an opportunity to make further revisions and to improve this manuscript.

Thanks again.

Reviewer 5 Report

Guangwan Zhang and co-workers report a ruthenium-doped anatase type VTi2.6O7.2 anode with high conductivity for enhanced sodium storage. They report improvement of performance of VTi2.6O7.2 as anode by Ru doping. The results are clear and may attract readers, and the authors revised the manuscript according to the reviewer’s comments. Although the authors showed the estimated crystallite size (not grain size) in the communication between the authors and the referee, they did not show in the manuscript. I think it is better to show it. From above reasons, the authors are advised to revise the manuscript.

Author Response

Response Letter

Manuscript ID: coatings-2193026

Ru-doped VTi2.6O7.2 anode with high conductivity for enhanced sodium storage

Guangwan Zhang1,2, Chunhua Han1,2*, Kang Han1, Jinshuai Liu1,2, Jinghui Chen1,2, Haokai Wang1,2, Lei Zhang, 2,3* Xuanpeng Wang2,3*

Response to Reviewer-#5:

We thank the Reviewer-#5 for the thoughtful and encouraging comments about our manuscript, and welcome the opportunity to address and clarify the issues raised in the report. Our responses to the points raised in the report are described below following specific Reviewer comments.

Comment:

Guangwan Zhang and co-workers report a ruthenium-doped anatase type VTi2.6O7.2 anode with high conductivity for enhanced sodium storage. They report improvement of performance of VTi2.6O7.2 as anode by Ru doping. The results are clear and may attract readers, and the authors revised the manuscript according to the reviewer’s comments. Although the authors showed the estimated crystallite size (not grain size) in the communication between the authors and the referee, they did not show in the manuscript. I think it is better to show it. From above reasons, the authors are advised to revise the manuscript.

Response to Comment:

We sincerely thank the Reviewer-#5's high remarks and important support of our work.

We have inserted a more detailed description for the crystallite size of Ru-VTO in the manuscript. As follows:

“We calculated the crystallite size of Ru-VTO to be 10.66 nm by applying the Scherrer equation (Equation S1) to the XRD profile [50].”

  1. Muniz, F.;T, ; Miranda, M, R.; Morilla dos Santos, C.; & Sasaki, J, M., The Scherrer equation and the dynamical theory of X-ray diffraction. Acta Crystallographica Section A: Foundations and Advances 2016, 72(3), 385-390.

The Scherrer equation and correlated descriptions are shown as Equation S1 in the supporting information 

Overall, we sincerely thank Reviewer-#5's kind and significant comments on our manuscript. These comments have provided us an opportunity to make further revisions and to improve this manuscript. We wish that these responses, additions and improvements will satisfy Reviewer-#5's comments. We believe that our revised manuscript has reached the standard for publication in Coatings.
